# Molecular Mechanisms of Pulmonary Fibrogenesis and Its Progression to Lung Cancer: A Review

**DOI:** 10.3390/ijms20061461

**Published:** 2019-03-22

**Authors:** Tomonari Kinoshita, Taichiro Goto

**Affiliations:** 1Division of General Thoracic Surgery, Department of Surgery, Keio University School of Medicine, 35 Shinanomachi, Shinjuku, Tokyo 1608582, Japan; kinotomo0415@gmail.com; 2Lung Cancer and Respiratory Disease Center, Yamanashi Central Hospital, Kofu, Yamanashi 4008506, Japan

**Keywords:** idiopathic pulmonary fibrosis, lung cancer, pathogenesis, common pathways

## Abstract

Idiopathic pulmonary fibrosis (IPF) is defined as a specific form of chronic, progressive fibrosing interstitial pneumonia of unknown cause, occurring primarily in older adults, and limited to the lungs. Despite the increasing research interest in the pathogenesis of IPF, unfavorable survival rates remain associated with this condition. Recently, novel therapeutic agents have been shown to control the progression of IPF. However, these drugs do not improve lung function and have not been tested prospectively in patients with IPF and coexisting lung cancer, which is a common comorbidity of IPF. Optimal management of patients with IPF and lung cancer requires understanding of pathogenic mechanisms and molecular pathways that are common to both diseases. This review article reflects the current state of knowledge regarding the pathogenesis of pulmonary fibrosis and summarizes the pathways that are common to IPF and lung cancer by focusing on the molecular mechanisms.

## 1. Introduction

Idiopathic pulmonary fibrosis is a progressive and usually fatal lung disease characterized by fibroblast proliferation and extracellular matrix remodeling, which results in irreversible distortion of the lung’s architecture. Although its cause remains to be elucidated fully, advances in cellular and molecular biology have greatly expanded our understanding of the biological processes involved in its initiation and progression [1]. It is widely accepted that environmental and occupational factors, smoking, viral infections, and traction injury to the peripheral lung can cause chronic damage to the alveolar epithelium [2]. Based on recent in vitro and in vivo studies of IPF, the novel therapeutic reagents pirfenidone and nintedanib were developed to slow the progression of this complex disease [3,4,5]. However, these drugs do not improve lung function and patients often remain with poor pulmonary function [6,7]. Furthermore, neither drug has been tested prospectively in patients with coexisting IPF and lung cancer [8]. In previous studies, 22% of patients with IPF developed primary lung cancers, corresponding with a five-fold greater risk than that in the general population [8,9,10,11,12]. Similarly, primary lung cancer risk is more than 20 times higher in patients who undergo lung transplantation for IPF than in the general population [13,14]. These observations warrant efforts to identify pathways that are common to both disorders. Questions regarding the proper and ideal management of patients who suffer from both IPF and lung cancer are also raised. It is assumed that pathogenetic similarities between IPF and lung cancer are a starting point for investigations of disease pathogenesis and the resulting insights will improve therapeutic approaches. This review article summarizes the current knowledge of the pathogenesis of pulmonary fibrosis and outlines the common molecular pathways between IPF and lung cancer.

## 2. The Pathogenesis of Pulmonary Fibrosis

Although knowledge of the pathogenesis of IPF remains incomprehensive, numerous research papers have contributed to the understanding of this disease. In particular, some environmental and microbial exposures have been associated with the initiation of IPF. Various individual genetic and epigenetic factors have also been related to the development of fibrosis, and potential contributions of variants and interactions with putative external factors have been presumed but not clarified. Repeated microinjury to alveolar epithelial tissues has been revealed as the first trigger of an aberrant repair process in which several lung cells develop abnormal behaviors that promote the fibrotic process. IPF is currently considered an epithelium-driven disease wherein dysfunctional aging lung epithelia are exposed to recurrent microinjuries that sabotage regeneration and lead to aberrant epithelial–mesenchymal crosstalk, creating an imbalance between profibrotic and antifibrotic mediators. Concurrently, environments that are supportive of elevated fibroblast and myofibroblast activities are maintained, and the normal repair mechanisms are replaced with chronic fibrosis. This review article details the current evidence of molecular contributions to the pathogenesis of IPF. The currently accepted mechanisms of pulmonary fibrosis are shown in Table 1 and Figure 1.

### 2.1. Dysfunctional Epithelia Trigger Aberrant Wound Healing Processes

It is assumed that fibrosis advances over long periods of time in patients with IPF. Thus, at the time of diagnosis, modifications of lung structure have already been established by the disease and pathological features, such as various stages of epithelial damage, alveolar epithelial cell (AEC) 2s hyperplasia, dense fibrosis, and abnormally proliferating mesenchymal cells, are found. At this time, it is not possible to determine the course of events that have led to lung damage; however, it is accepted that dysfunctional epithelia are key to the pathogenesis of IPF [15].

Under normal conditions of lung injury, AEC1s are replaced with proliferating and differentiating AEC2 cells and stem cells, which restore alveolar integrity by stimulating coagulation, the formation of new vessels, activation and migration of fibroblasts, and synthesis and proper alignment of collagen. Chemokines, such as transforming growth factor (TGF)-β1, platelet-derived growth factor (PDGF), vascular endothelial growth factor (VEGF), and fibroblast growth factor (FGF), are central to these processes. Conversely, continued lung injury or loss of normal restorative capacity invokes an inflammatory phase of the wound healing process. The associated increases in the expression levels of interleukin-1 (IL-1) and tumor necrosis factor-alpha (TNF-α) create a biochemical environment that favors chronic flaws of regeneration and tissue remodeling [16].

### 2.2. Growth Factors Associated with the Initial Stages of Pulmonary Fibrogenesis

#### 2.2.1. TGF-β

TGF-βs are multifunctional cytokines that are present as three isoforms: TGF-β1, TGF-β2, and TGF-β3. Although the biological activities of these isoforms are indiscrete, TGF-β1 plays a predominant role in pulmonary fibrosis [17]. The three TGF-β receptors, type I (TGFRI), type II (TGFRII), and type III (TGFRIII), have the potential to bind to all three TGF-βs with high affinity. However, TGF-β is the best characterized promoter of extracellular matrix (ECM) production and is considered the strongest chemotactic factor for immune cells, such as monocytes and macrophages. In these cell types, TGF-β activates the release of cytokines, such as PDGF, IL-1β, basic FGF (bFGF), and TNF-α, and autoregulates its own expression. Increases in TGF-β production are consistently observed in epithelial cells and macrophages from lung tissues of patients with IPF [18] and in rodents with bleomycin-induced pulmonary fibrosis [19]. Smad proteins are known as mediators of TGF-β signaling from the membrane to the nucleus [20]. Activated TGF-β receptors induce phosphorylation of Smad2 and Smad3, and complexes of these with other Smad proteins are translocated into the nucleus to regulate transcriptional responses. Studies show that the deficiency of Smad3 attenuates bleomycin-induced pulmonary fibrosis in mice [21] and that the inhibitory Smad7 prevents the phosphorylation of Smad2 and Smad3 via activated TGF-β receptors [22,23].

TGF-β1 is considered the most important mediator of IPF. AEC2s produce TGF-β1 following actin–myosin-mediated cytoskeletal contractions that are induced by the unfolded protein response (UPR) following ανβ6 integrin activation. The αvβ6 integrin/TGF-β1 pathway is a constitutively expressed molecular sensing mechanism that is primed to recognize injurious stimuli. TGF-β1 is a strong profibrotic mediator that promotes the epithelial–mesenchymal transition (EMT); epithelial cell apoptosis; epithelial cell migration; other profibrotic mediator production; circulating fibrocyte recruitment; fibroblast activation and proliferation and transformation into myofibroblasts; and VEGF, connective-tissue growth factor, and other pro-angiogenic mediator production [24].

#### 2.2.2. PDGF

PDGF is a potent chemoattractant for mesenchymal cells and induces the proliferation of fibroblasts and the synthesis of ECM. Activated homologous A and B subunits of PDGF can form three dimeric PDGF isoforms. Alveolar macrophages with IPF produce higher volumes of PDGF-B mRNA and protein [25,26]. AEC2s and mesenchymal cells also express abnormal levels of PDGF in animal models [27]. Moreover, PDGF-B transgenic mice develop lung disease with diffusely emphysematous lung lesions and inflammation/fibrosis in focal areas [28]. In agreement, intratracheal instillation of recombinant human PDGF-B into rats produces fibrotic lesions that are concentrated around large airways and blood vessels [29]. In another study, gene transfer of an extracellular domain of the PDGF receptor ameliorated bleomycin-induced pulmonary fibrosis in a mouse model [30]. Insulin-like growth factor (IGF)-1 also promoted fibroblast proliferation synergistically with PGDF [31]. Accordingly, alveolar macrophages from patients with IPF expressed IGF-1 mRNA and protein at greater levels than those in normal alveolar macrophages [31,32].

#### 2.2.3. FGF

bFGF is a stimulator of fibroblast and endothelial cell proliferation that has been correlated with the proliferative aspects of fibrosis. In particular, bFGF expression is up-regulated at various periods of wound healing, and recombinant bFGF has been shown to accelerate wound healing. Accordingly, anti-bFGF antibody inhibited the formation of granulated tissue and normal wound repair. Alveolar macrophages are a predominant source of bFGF in intra-alveolar fibrotic areas following acute lung injury [33]. In a study of IPF, mast cells were found to be the predominant bFGF-producing cells, and bFGF levels were associated with bronchoalveolar lavage cellularity and with the severity of gas exchange abnormalities [34].

#### 2.2.4. TGF-α

TGF-α induces proliferation in endothelial cells, epithelial cells, and fibroblasts, and is present in fibrotic areas [35]. In proliferative fibrotic lesions in rats with asbestos- or bleomycin-induced pulmonary fibrosis, AECs and macrophages had elevated expression levels of TGF-α [36]. Similarly, in transgenic mice expressing human TGF-α, proliferative fibrotic responses in interstitial and pleural surfaces were epithelial cell specific [37]. These results indicate that TGF-α is involved in cell proliferation under fibrotic conditions following lung injury.

#### 2.2.5. Keratinocyte Growth Factor (KGF)

KGF is produced by mesenchymal cells, and the KGF receptor is expressed in the epithelial tissues of developing lungs. In rats, KGF accelerated the functional differentiation of AEC2s, and the intratracheal instillation of KGF significantly improved bleomycin-induced pulmonary fibrosis [38]. These data suggest that KGF participates in the maintenance and repair of alveolar epithelium and has potential in the treatment of lung injury and pulmonary fibrosis.

#### 2.2.6. Hepatocyte Growth Factor (HGF)

HGF is produced by mesenchymal cells and has been identified as a potent mitogen for mature hepatocytes. The HGF receptor is a c-Met proto-oncogene product that is predominantly expressed in various types of epithelial cells. HGF levels are higher in bronchoalveolar lavage fluid and serum from patients with IPF than in serum from healthy people [39,40]. HGF is also highly expressed by hyperplastic AECs and macrophages in lung tissues of patients with IPF. In in vitro studies of epithelial cells, HGF promoted DNA synthesis in AEC2s [41]. The administration of HGF also inhibited fibrotic changes in mice with bleomycin-induced lung injury [42]. Promisingly, the combination of HGF and interferon-γ (IFN-γ) enhanced the migratory activity of A549 cells by up-regulating the c-Met/HGF receptor [43]. Based on these observations, HGF treatments may offer a novel strategy for promoting the repair of inflammatory lung damage for patients with pulmonary fibrosis.

### 2.3. Changes in AEC2s that Lead to Aberrant Tissue Repair

Repetitive exposures of alveolar epithelium to microinjuries, such as infection, smoking, toxic environmental inhalants, and gastroesophageal reflux, contribute to AEC1 damage. AEC2s normally regenerate damaged cells, but when dysfunctional, their ability to reestablish homeostasis is impaired. This condition is considered indicative of the pathogenesis of IPF [44,45].

#### 2.3.1. UPR

High cellular activity leads to protein over-expression, and if unchecked, it can cause endoplasmic reticulum (ER) stress. The correcting protective pathway is stimulated by the imbalance between cellular demand for protein synthesis and the capacity of the ER to dispose of unfolded or damaged proteins. This protective pathway is known as UPR, and it re-establishes ER homeostasis. To this end, this pathway inhibits protein translation, targets proteins for degradation, and induces apoptosis when overwhelmed. The activation of UPR stimulates the expression of profibrotic mediators, such as TGF-β1, PDGF, C-X-C motif chemokine 12 (CXCL12), and chemokine C-C motif ligand 2 (CCL2), and thus, can lead to apoptosis [46].

#### 2.3.2. Epithelial–Mesenchymal Transition (EMT)

EMT is a molecular reprograming process, and in AEC2s, it is induced by UPR and enhanced by profibrotic mediators and signaling pathways. Under these conditions, epithelial cells express mesenchymal cell-associated genes, detach from basement membranes, and migrate and down-regulate their typical markers. The most used marker of these transitioning cells is alpha smooth-muscle actin (αSMA). However, EMT occurs during development and in cancerous and fibrotic tissues, but it is not involved in the restoration of tissues through wound healing processes [46].

#### 2.3.3. Wnt-β-Catenin Signaling

Other key pathways of IPF are related to the deregulation of embryological programs, such as Wnt-β-catenin signaling, which has been associated with EMT and fibrogenesis following activation by TGF-β1, sonic hedgehog, gremlin-1, and phosphatase and tensin homolog. Deregulation of these pathways confers resistance to apoptosis and offers proliferative advantages to cells [47].

### 2.4. Endothelium and Coagulation

Damage to alveolar structures and the loss of AECs with basement membranes involves alveolar vessels and leads to increased vascular permeability. Wound clots form during this early phase of wound healing responses, and sequentially, new vessels are formed through the proliferation of endothelial cells and endothelial progenitor cells (EPCs). Patients with IPF with failure of re-endothelization have significantly decreased numbers of EPCs, likely resulting in dysfunctional alveolar–capillary barriers, profibrotic responses, and compensatively augmented VEGF expression. This series of endothelial changes could stimulate fibrotic processes and abnormalities of vessel functions, contributing to cardio–respiratory declines and advanced disease. Furthermore, endothelial cells may undergo a mesenchymal transition with similar consequences as those of EMT [48].

Endothelial and epithelial damage also activates coagulation cascades during the early phases of wound healing. Coagulation proteinases have several cellular effects on wound healing. In particular, the tissue factor-dependent pathway is central to the pathogenesis of IPF and promotes a pro-coagulation state with increased levels of inhibitors of plasminogen activation, active fibrinolysis, and protein C. Under these pro-coagulation conditions, degradation of ECM is decreased, resulting in profibrotic effects and the induction of fibroblast differentiation into myofibroblasts via proteinase-activated receptors [16].

### 2.5. Immunogenic Changes that Lead to Pulmonary Fibrosis

The pathobiology of IPF is led by aberrant epithelial–mesenchymal signaling, but inflammation may also play an important role because inflammatory cells are involved in normal wound healing from early phases. Initially, macrophages produce cytokines that induce inflammatory responses and participate in the transition to healing environments by recruiting fibroblasts, epithelial cells, and endothelial cells. If injury persists, neutrophils and monocytes are recruited, and the production of reactive oxygen species exacerbates epithelial damage. The resulting imbalances between antioxidants and pro-oxidants may also promote apoptosis of epithelial cells and activation of pathways that impair function. Finally, monocytes and macrophages produce PDGF, CCL2, macrophage colony stimulating factor, and colony stimulating factor 1. These proteins may also have direct profibrotic effects [44,49].

The roles of lymphocytes in IPF are still unclear. However, some lymphocytic cytokines are considered profibrotic due to their direct effects on the activities of fibroblast and myofibroblast. Th-1, Th-2, and Th-17 T-cells have been clearly associated with the pathogenesis of IPF. The Th1 T-cell subset produces IL-1α, TNF-α, PDGF, and TGF-β1 and has net profibrotic effects. Th2 and Th17 responses appear more important in the pathogenesis of IPF. In particular, the typical Th2 interleukin IL-4 induces IL-5, IL-13, and TGF-β1 expression, leading to the recruitment of macrophages, mast cells, eosinophils, and mesenchymal cells and the direct activation of fibroblasts. Additionally, fibroblasts from patients with IPF are hyperresponsive to IL-13, which has a positive effect on fibroblast activity and enhances the production of ECM. The Th17 T-cell subset indirectly promotes fibrosis by increasing TGF-β1 levels. Th17 cells are also positively regulated by TGF-β1, suggesting the presence of a positive feedback loop [16]. Numbers of regulatory T-cells are reportedly lower in bronchoalveolar lavage fluid and peripheral blood samples from patients with IPF than in those of healthy subjects. Regulatory T-cells (Tregs) play a crucial role in immune tolerance and the prevention of autoimmunity; deficiencies in numbers and functions of these T-cells play an important role in the initial phases of pathogenesis of IPF. The function of Treg in IPF is severely impaired due to reduced number of infiltrating Tregs in addition to dysfunction of Tregs. Interestingly, the compromised Treg function in bronchoalveolar lavage is associated with parameters of the disease severity of IPF, indicating a causal relationship between the development of IPF and impaired immune regulation mediated by Tregs [50]. Previous studies have demonstrated low IFN-γ levels in the lungs of patients with IPF. IFN-γ inhibits fibroblastic activity and abolishes Th2 responses. However, further studies are required to characterize the roles of inflammation in the pathobiology of IPF. Currently, the early stages of IPF are poorly understood, as are the mechanisms of disease progression [49,51]. Nonetheless, pirfenidone (5-methyl-1-phenyl-2-[1H]-pyridone) was designed to have anti-inflammatory and antifibrotic effects and was efficacious in the clinical setting [6].

### 2.6. Interactions Between ECM and Mesenchymal Cells, Fibrocytes, Fibroblasts, and Myofibroblasts

Contributions of mesenchymal cells, and particularly fibroblasts and myofibroblasts, are crucial for the pathogenesis of IPF. These cells are recruited, activated, and induced to differentiate and proliferate in the abnormal biochemical environments that are created by activated epithelial and endothelial cells. Although the initial trigger and source of mesenchymal cell recruitment remain unclear, the current published consensus defines fibroblasts and myofibroblasts as the key cell types for IPF. Circulating fibrocytes, pulmonary fibroblasts, and myofibroblasts have also been identified among mesenchymal cells that are involved in IPF [52]. The most recent studies of these processes are summarized in a well-integrated review [53].

## 3. Common Characteristics of IPF and Lung Cancer

Multiple studies compare IPF with cancer to provide insights into the pathogenesis of both diseases, for which survival rates are low. Arguments against the similarities of cancer and IPF include the presence of homogeneity, metastases, and laterality in cancers. However, cytogenetic heterogeneity has been shown in myofibroblasts, which do not metastasize to other organs. In addition, simultaneous involvement of both lungs is a definitive indication of IPF. However, this is primarily based on the generally accepted assumption that tumors are almost always monoclonal and grow in only one lung before metastasizing and invading other organs. From an anatomical viewpoint, patients with IPF mainly exhibit fibrosis in the lung periphery and in the lower lobes, which are sites of lung tumors in a high percentage of cases [54]. Additionally, patients with lung transplants due to IPF have much higher rates of lung cancer, as stated above [13,14]. These observations warrant further studies regarding the molecular connections between these two lung diseases. Furthermore, epigenetic and genetic abnormalities, changed relationships between cells, uncontrolled proliferation, and abnormal activation of specific signal transduction pathways are pathogenic features of both diseases [55,56]. Principal fibrogenic molecules, signal transduction pathways and immune cells that potentially participate both in two diseases are shown in Table 2.

### 3.1. Epigenetic and Genetic Abnormalities

Hypomethylation of oncogenes and methylation of tumor suppressor genes are established pathogenic mechanisms for most tumors. Epigenetic responses to environmental exposures, including smoking and dietary factors, and aging have recently been identified in patients with IPF. Recent studies also demonstrated changes to global methylation patterns in patients with IPF that are reciprocal to those in patients with lung cancers [57]. Under the conditions of IPF, hypermethylation of the CD90/Thy-1 promoter region decreases the expression of the glycoprotein Thy-1, which is normally expressed by fibroblasts [58,59]. The loss of this molecule in patients with IPF also correlates with invasive behaviors of cancers and the transition from fibroblasts into myofibroblasts. Hence, pharmaceutical inhibition of the methylation of *Thy-1* gene may restore Thy-1 expression, suggesting a new therapeutic approach for this disease. Specific gene mutations have also been considered important to the origin and progression of cancer [60]. Similarly, expression of the oncogene p53, fragile histidine triads, microsatellite instability, and loss of heterozygosity were observed in approximately half of the cases of IPF, frequently in the peripheral honeycombed lung regions that are specifically characteristic of IPF [60,61,62,63]. Additionally, mutations that are generally related to cancer occurrence and development, including those affecting telomere shortening and telomerase expression, have been observed in familial IPF [64,65,66]. Recently, circulating and cell-free DNA has been considered as a diagnostic and prognostic biomarker of cancer [67]. In these studies, free circulating concentrations of DNA increased in patients with cancer and IPF compared with that in patients with other fibrotic lung diseases [68]. In addition to circulating DNA, abnormal expression levels of mRNA were correlated with the pathogenesis of both diseases. These studies suggest that short non-protein-coding RNAs regulate carcinogenesis related genes that are involved in growth, invasion, and metastasis; these features are characteristic of cancer cells [69,70,71]. Recent papers show that 10% of mRNAs are aberrantly expressed in patients with IPF [72,73,74]. Among them, let-7, miR-29, miR-30, and miR-200 were down-regulated, whereas miR-21 and miR-155 were up-regulated. These changes corresponded with groups of genes that are associated with fibrosis, regulation of ECM, induction of EMT and apoptosis. Some of these mRNAs may also affect and be affected by TGF-β expression, potentially speeding functional deterioration in patients with IPF.

### 3.2. Abnormal Cell–Cell Communication

Intercellular channels provide metabolic and electrical coupling of cells and are formed by proteins of the connexins (Cxs) family. Cxs are necessary for the synchronization of cell proliferation and tissue repair [75]. Among them, Cx43 is the most abundant on fibroblast membranes and is involved in tissue repair and wound healing. At wound sites, the repression of Cx43 promotes repair of injured skin tissues with increased cell proliferation and migration of keratinocytes and fibroblasts. Accordingly, down-regulation of Cx43 is related to increased expression levels of TGF-β and production of collagen and acceleration of the differentiation of myofibroblast, which likely promotes healing. These changes contribute to the loss of control over the proliferation of fibroblasts that characterizes abnormal repair and fibrosis. This contention is supported by observations of low expression of Cx43 in fibroblasts derived from keloids and hypertrophic scars than in those derived from normal skin tissues [76]. Although low expression levels of Cxs are often correlated with the progression of cancer and the loss of intercellular communication [77], human lung carcinoma cell lines with high expression of Cx43 showed reduced proliferation [78]. Reduced expression of Cx43 was reported in primary lung fibroblasts from patients with IPF, and reduced intercellular communication was also identified in these cells [79]. Limited cell–cell communications are often reported in fibroblasts from patients with IPF and in cancer cells, reflecting common defects of contact inhibition and uncontrolled proliferation.

### 3.3. Abnormal Activation of Signaling Pathways

The Wnt/β-catenin signaling pathway regulates molecules that are related to tissue invasion, such as matrilysin, laminin, and cyclin-D1. However, arguably, the most important function of Wnt/β-catenin pathway is to mediate crosstalk with TGF-β. This pathway is abnormally activated in some tumors, as shown in lung cancer and mesothelioma [80]. Wnt/β-catenin pathway activation was also shown recently in fibroproliferative disorders of liver and kidney tissues [81]. The Wnt/β-catenin pathway is strongly activated in the lung tissues of patients with IPF [82], potentially reflecting the activities of TGF-β [83]. Specifically, TGF-β potentially activates extracellular signal-regulated protein kinases 1 and 2 (ERK1/2), and the target genes of this pathway activate other signaling pathways, including the phosphatidylinositol 3-kinase (PI3K)/Akt pathway, which regulates proliferation and apoptosis. The roles of PI3K in proliferation and differentiation into myofibroblasts have been demonstrated following stimulation with TGF-β [84]. In cancer cells, the activation of PI3K pathway participates in the demise of regulatory controls over cell proliferation. Therapeutic inhibitors have been developed using the PI3K pathway as a target, and their effects on tumor growth and survival is being assessed in many cancers [85]. Oral administration of a PI3K pathway inhibitors significantly prevented bleomycin-induced pulmonary fibrosis in rats [86]. Hence, clinical trials of such inhibitors are eagerly awaited for patients with IPF.

Tyrosine kinases are key mediators of multiple signaling pathways in healthy cells with demonstrated roles in cell growth, differentiation, adhesion, and motility and in the regulation of cell death. Tyrosine kinase activity is controlled by specific transmembrane receptors that mediate the activity of various ligands. Conversely, abnormal activities of these kinases have been associated with development, progression, and spread of several types of cancer [87]. Recently, activities of tyrosine kinase receptors were investigated in wound healing process and fibrogenesis.

TGF-β, PDGF, VEGF, and FGF are common mediators of carcinogenesis and fibrogenesis. Among them, VEGF may directly or indirectly promote cell survival and proliferation by activating ERK1/2 and PI3K. Accordingly, elevated expression levels of VEGF mRNA were shown in EPCs from patients with IPF. Furthermore, antifibrotic strategies using multiple inhibitors of tyrosine kinase receptors have been evaluated in a rat model of bleomycin-induced fibrosis; PDGF, VEGF, and FGF inhibitors produced significant improvement in fibrosis [48,88,89,90]. In support of these in vitro and in vivo observations, the multiple tyrosine kinase inhibitor nintedanib showed highly favorable results for the treatment of IPF [7].

### 3.4. Abnormal Migration and Invasion Activities

TGF-β is the most important mediator of the pathogenesis and carcinogenesis of IPF. In tumor microenvironments, TGF-β, predominantly from cancer-derived epithelial cells, induces myofibroblast recruitment at the invasive front of the cancer tissue and protects myofibroblasts from apoptosis. These cells encircle tumor tissues and produce TGF-β. With inflammatory mediators and metalloproteinases, myofibroblasts break basement membranes of surrounding tissues to facilitate tumor invasion [91,92]. Likewise, in IPF, myofibroblasts maintain proliferation through autocrine production of TGF-β, leading to their uncontrolled proliferation [93]. Moreover, related, antifibrotic prostaglandin E2 is down-regulated in myofibroblasts from IPF tissues [94]. TGF-β1 promotes the nuclear localization of myocardin-related transcription factor-A (MRTF-A), which regulates the differentiation and survival of fibroblasts, resulting in enhanced lung fibrosis [95,96,97,98]. MRTF-A has been targeted as a mediator of tumor progression and metastasis [99,100,101].

In cancer cells, the capacity to invade surrounding tissue strongly correlates with the expression of various molecules, including laminin, heat shock protein 27, and fascin [102,103,104]. In IPF, epithelial cells around fibroblast foci also express these molecules [105]. However, these molecules are exclusively expressed by bronchiolar basal cells, which are located as a layer between luminal epithelial cell and myofibroblast layers. Hence, these molecules are likely contributors to the migration of cells and the invasion of bronchiolar basal cells into myofibroblasts and luminal epithelium and are expressed at the invasive front of tumors.

Matrix metalloproteases and integrins are strongly associated with invasion and migration of cells [106]. Integrins activate cancer cells through the KRAS/RelB/NF-κB pathway and lead to the development of stem cell-like properties, such as independent growth and drug resistance. These properties provide cell–cell communications between inflammatory cells, fibroblasts, and parenchymal cells through ECM. Under conditions of IPF, integrin promotes initiation, maintenance, and resolution of tissue fibrosis. Accordingly, integrin expression was reportedly high in myofibroblasts and AECs after lung injury. Integrin is also considered a strong regulator of TGF-β during the progression of lung fibrosis. A clinical study of the humanized antibody STX-100 has been conducted for IPF [107]. Other inhibitors, such as specific antibodies against αvβ6, have also been investigated in clinical trials, and these antibodies were tested in preclinical models of fibrosis and in the murine model of bleomycin-induced pulmonary fibrosis.

### 3.5. Inflammatory Environment

Inflammatory reaction is described by some reports as a promoting factor in the development and progression step of tumorigenesis [108]. As described above, some kinds of macrophages produce cytokines which contribute to the inflammatory responses such as fibrosis-associated macrophages. This macrophage behaves as an M2 phenotype macrophage expressing arginase and CD206 [109]. M2 macrophages have been broadly identified as trigger cells towards tumor progression [110,111,112]. Myeloid-derived suppressor cells are associated with poor prognosis in malignancies and their accumulation in IPF is also correlated with disease progression [113]. On the other hand, infiltrating T lymphocytes play a crucial role in tumor progression and suppression, although their roles in IPF are still unclear [114]. Infiltrating Tregs are significantly correlated with the tumor progression whereas deficiency in numbers and functions of Tregs is observed in the initial step of IPF (Table 2) [50,115]. Further studies regarding the role of Treg in the IPF-related cancer are awaited.

## 4. Conclusions

In conclusion, cancer and fibrosis are both severe lung diseases, and they share biological pathways. Although the specific genetic and cellular mechanisms are not yet fully understood, several signaling pathways and microenvironments have been shown to disrupt tissue architecture and lead to dysfunction. Conversely, it is clear that lung tumorigenesis and fibrosis display highly heterogeneous behaviors, warranting personalized therapeutic approaches. Lung fibrosis may eventually be attenuated by therapies that are developed after considering mechanisms that are common to cancer and IPF.

## Figures and Tables

**Figure 1 ijms-20-01461-f001:**
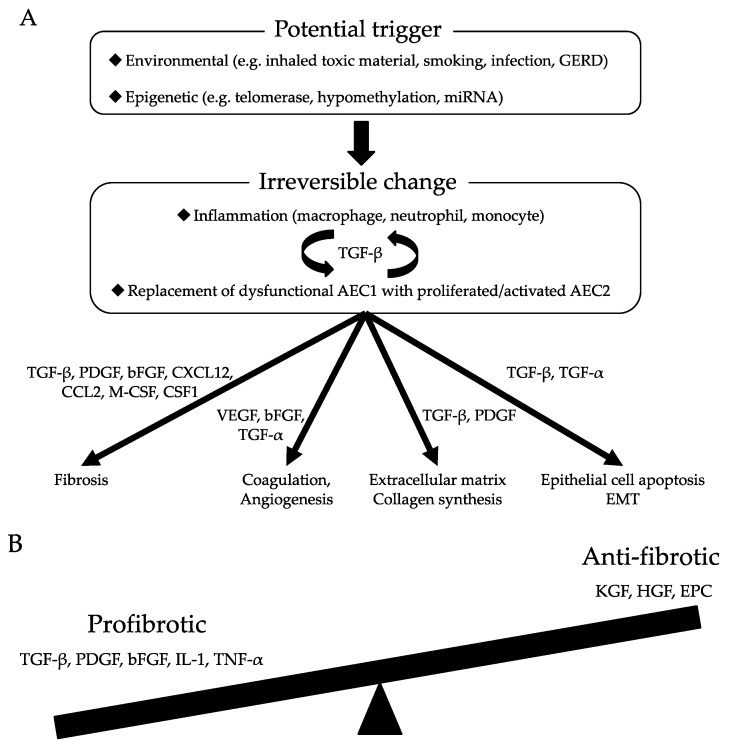
(**A**) Molecular mechanisms of pulmonary fibrosis; (**B**) Imbalance of profibrotic and antifibrotic mediators lead to defective regeneration and aberrant remodeling, resulting in the pathological transformation of pulmonary fibrosis. GERD, gastroesophageal reflux disease; AEC, alveolar epithelial cell; EMT, epithelial mesenchymal transition; EPC, Endothelial progenitor cell.

**Table 1 ijms-20-01461-t001:** Functions of representative molecules contributing to idiopathic pulmonary fibrosis.

Molecules	Profibrotic/Antifibrotic	Function in IPF
TGF-β	Profibrotic	Extracellular matrix production
		Epithelial mesenchymal transition
		Epithelial cell apoptosis and migration
		Recruitment of fibrocytes and immune cells
		Fibroblast activation, myofibroblast proliferation
		Induction of growth factor production
		Induction of pro-angiogenic mediator production
PDGF	Profibrotic	Extracellular matrix production
		Fibroblast proliferation
FGF	Profibrotic	Fibroblast activation
		Endothelial cell proliferation
TGF-α	Profibrotic	Epithelial cell proliferation
		Endothelial cell proliferation
		Fibroblast proliferation
VEGF	Profibrotic	Angiogenesis in injured lung
KGF	Antifibrotic	Maintenance and repair of injured lung
HGF	Antifibrotic	Maintenance and repair of injured lung

**Table 2 ijms-20-01461-t002:** Principal factors participating both in lung cancer and idiopathic pulmonary fibrosis.

Mediators		IPF	Lung Cancer
Abnormal mRNA	let-7	down-regulated	down-regulated
	miR-21	up-regulated	up-regulated
	miR-29	down-regulated	down-regulated
	miR-30	down-regulated	down-regulated
	miR-155	up-regulated	up-regulated
	miR-200	down-regulated	down-regulated
Cell-free DNA	-	up-regulated	up-regulated
Glycoprotein	Thy-1	down-regulated	down-regulated
Connexin	Cx43	down-regulated	down-regulated
Growth Factors	TGF-β	up-regulated	up-regulated
	PDGF	up-regulated	up-regulated
Migration	VEGF	up-regulated	up-regulated
	FGF	up-regulated	up-regulated
	laminin	up-regulated	up-regulated
	fascin	up-regulated	up-regulated
Pathways	heat shock protein 27	up-regulated	up-regulated
	Wnt pathway	up-regulated	up-regulated
	PI3K/Akt pathway	up-regulated	up-regulated
Immune Cells	FAM	up-regulated	up-regulated
	MDSC	up-regulated	up-regulated
	Treg	down-regulated	up-regulated

FAM, fibrosis-associated macrophage. MDSC, myeloid-derived suppressor cell. Treg, regulatory T-cell.

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
