# Peer review of "Molecular Mechanisms of Pulmonary Fibrogenesis and Its Progression to Lung Cancer: A Review"

_ijms, 2019, doi:10.3390/ijms20061461_

Round 1
Reviewer 1 Report
In this review, Tomonari Kinoshita and Taichiro Goto point out important pathways contributing to idiopathic pulmonary fibrosis (IPF) and describe potential links that may favor the development of lung cancer. As a whole, this review is well organized and provides sufficient amount of detail. However, I would recommend the following points to be addressed before publication.
1) The first sentence of the abstract and the first sentence of the introduction are too vague and need rewriting.
2) It would be valuable to add a table recapitulating “pro-fibrotic” and “pro- cancer” factors discussed in this review, highlighting possible link between IPF and lung cancer whenever possible.
3) In line with section 2.5, It could be important to add a section describing how the immunological environment observed in IPF patients could contribute to the emergence of cancer. In particular, the role of regulatory T cells can be addressed in this context.
Minor points
1) Line 295 “This pathway is abnormally activate in some tumors”. Please correct by “This pathway is abnormally activated in some tumors”.
2) Line 328. The sentence starting with “Although uncontrolled proliferation of myofibroblasts” is unclear. Please improve.
3) Line 32: Please verify that “often leave patients” is suitable. Otherwise it can be changed for instance by “However, these drugs do not improve lung function and patients remain with poor pulmonary function”.
Author Response
1) The first sentence of the abstract and the first sentence of the introduction are too vague and need rewriting.
Response: We replaced the sentences with new and clear descriptions.
2) It would be valuable to add a table recapitulating “pro-fibrotic” and “pro- cancer” factors discussed in this review, highlighting possible link between IPF and lung cancer whenever possible.
Response: We added Table 2 in section 3, and recapitulated the factors participating both in lung cancer and IPF.
3) In line with section 2.5, It could be important to add a section describing how the immunological environment observed in IPF patients could contribute to the emergence of cancer. In particular, the role of regulatory T cells can be addressed in this context.
Response: We agree with the reviewer. Accordingly, we put some descriptions in section 2.5, as follows.
The function of Treg in IPF is severely impaired due to reduced number of infiltrating Tregs in addition to dysfunction of Tregs. Interestingly, the compromised Treg function in bronchoalveolar lavage is associated with parameters of the disease severity of IPF, indicating a causal relationship between the development of IPF and impaired immune regulation mediated by Tregs.
Furthermore, we put a new section (section 3.5.) and addressed the inflammatory environment contributing to the emergence of cancer, as follows.
3.5. Inflammatory environment
Inflammatory reaction is described by some reports as a promoting factor in the development and progression step of tumorigenesis [108]. As described above, some kinds of macrophages produce cytokines which contribute to the inflammatory responses such as fibrosis-associated macrophages. This macrophage behaves as an M2 phenotype macrophage expressing arginase and CD206 [109]. M2 macrophages have been broadly identified as trigger cells towards tumor progression [110-112]. Myeloid-derived suppressor cells are associated with poor prognosis in malignancies and their accumulation in IPF is also correlated with disease progression [113]. On the other hand, infiltrating T lymphocytes play a crucial role in tumor progression and suppression, although their roles in IPF are still unclear [114]. Infiltrating Tregs are significantly correlated with the tumor progression whereas deficiency in numbers and functions of Tregs is observed in the initial step of IPF [50,115]. Further studies regarding the role of Treg in the IPF-related cancer are awaited.
Minor points
1) Line 295 “This pathway is abnormally activate in some tumors”. Please correct by “This pathway is abnormally activated in some tumors”.
Response: We corrected our mistake.
2) Line 328. The sentence starting with “Although uncontrolled proliferation of myofibroblasts” is unclear. Please improve.
Response: The sentence was replaced with a meaningful one.
3) Line 32: Please verify that “often leave patients” is suitable. Otherwise it can be changed for instance by “However, these drugs do not improve lung function and patients remain with poor pulmonary function”.
Response: We revised the sentence according to the reviewer’s suggestion.
Thank you very much for your thoughtful suggestions.
Reviewer 2 Report
Overall, this review article describing factors promoting Idiopathic lung fibrosis is well written. It does encompass the recent and established players in IPF development and in my opinion, worthy of publication.
I have a few minor suggestions which the authors may choose to incorporate:
1) It is recommended that the authors decipher each factor (such as TGF, vegf etc) in a table or chart showing the exact phenotype in IPF.
2) Can the authors summarize the remaining key unanswered questions for the field in tabular form?
Author Response
Reviewer 2
I have a few minor suggestions which the authors may choose to incorporate:
1)It is recommended that the authors decipher each factor (such as TGF, vegf etc) in a table or chart showing the exact phenotype in IPF.
Response: We added Table 1 and added some explanations regarding the functions of representative molecules affecting the progression of idiopathic pulmonary fibrosis.
2) Can the authors summarize the remaining key unanswered questions for the field in tabular form?
Response: This review article summarizes the current knowledge of the pathogenesis of pulmonary fibrosis and outlines the common molecular pathways between IPF and lung cancer. The specific genetic and cellular mechanisms are not yet fully understood, and in particular, further studies regarding the role of Treg in the IPF-related cancer are awaited.
We added these descriptions in the revised manuscript.
Thank you very much for your thoughtful suggestions.